# Learning without Global Backpropagation via Synergistic Information Distillation

## Abstract

Backpropagation (BP), while foundational to deep learning, imposes two critical scalability bottlenecks: **update locking**, where network modules remain idle until the entire backward pass completes, and **high memory consumption** due to storing activations for gradient computation. To address these limitations, we introduce *Synergistic Information Distillation* (SID), a novel training framework that reframes deep learning as a cascade of local cooperative refinement problems. In SID, a deep network is structured as a pipeline of modules, each imposed with a local objective to refine a probabilistic "belief" about the ground-truth target. This objective balances fidelity to the target with consistency to the belief from its preceding module. By decoupling the backward dependencies between modules, SID enables parallel training and hence eliminates update locking and drastically reduces memory requirements. Meanwhile, this design preserves the standard feed-forward inference pass, making SID a versatile drop-in replacement for BP. We provide a theoretical foundation, proving that SID guarantees monotonic performance improvement with network depth. Empirically, SID consistently matches or surpasses the classification accuracy of BP, exhibiting superior scalability and pronounced robustness to label noise. The code is publicly available at: `https://anonymous.4open.science/r/sid_BDEF`.

## 1 Introduction

The backpropagation algorithm (BP) is the cornerstone of deep learning, enabling efficient credit assignment through gradient-based optimization. Its mechanism relies on the chain rule to compute gradients via a sequential end-to-end traversal of the network's computation graph: a forward pass to compute activations and the final loss, followed by a backward pass to propagate the loss gradient from the output back to the input. While remarkably effective, this architectural design imposes two fundamental scalability bottlenecks:

- **Update Locking:** The sequential nature of the backward pass creates a global dependency. Parameters in a given layer cannot be updated until all subsequent layers have completed their backward computation. This locks the entire network during the gradient calculation, preventing parallel updates across layers and leading to significant computational idling.
- **High Memory Consumption:** To compute the local gradients at each layer, the corresponding activations from the forward pass must be stored in memory. For deep networks, the aggregate memory required to store these activations for the entire graph becomes a primary constraint, limiting model depth and training batch size.

These challenges have motivated a rich body of research into backpropagation-free alternatives. However, existing approaches often introduce their own significant trade-offs. *Target Propagation* methods (Lee et al., 2015; Ernoult et al., 2022) require complex auxiliary networks to learn approximate inverses, which can be difficult to train and unstable. *Equilibrium-based models* (Scellier & Bengio, 2017) replace the standard forward pass with iterative dynamics to reach a fixed point, fundamentally altering the inference process and its computational cost. While other methods like *Feedback Alignment* (Nøkland, 2016) address related issues such as biological plausibility, they do not resolve the core bottlenecks of update locking or memory overhead (Sun et al., 2025; Rivaud et al., 2025; Somasundaram et al., 2025; Li et al., 2025; Chen et al., 2025; Aghagolzadeh & Ezoji, 2025; Ren & Li, 2024; Gong et al., 2025; Feng et al., 2024).

In this paper, we introduce *Synergistic Information Distillation* (SID), a new paradigm that resolves BP's core limitations without the aforementioned trade-offs. Instead of approximating global gradients, SID reframes learning as a cooperative sequential refinement of a probabilistic belief. We view a network as a pipeline of modules, where each module $f_i$ receives a belief distribution $p_{i-1}$ from its predecessor and outputs a refined belief $p_i$. To achieve this, each module is imposed with a simple local objective composed of two terms: a distillation term that pulls the belief towards the ground-truth label and a consistency term that regularizes the refinement step, ensuring that the belief of the given module does not deviate drastically from the belief of the previous module.

The key to SID's efficacy is its mechanism for decoupling modules. By applying a stop-gradient operation to the consistency term, we decouple all backward dependencies between modules. This simple yet powerful design directly resolves the core limitations of BP:

1. **Update Locking is Eliminated.** With no inter-module dependencies, all modules can compute their gradients and be updated in parallel after a single, gradient-free forward pass that generates "teacher" beliefs.
2. **Memory Overhead is Reduced.** Since gradients are local to each module, only the activations within a single module need to be stored at any given time, leading to a memory footprint that scales with the size of the largest module, not the entire network depth.

Meanwhile, this design preserves a standard feed-forward architecture for inference, making it a seamless replacement for BP during training. Our contributions are summarized as follows:

1. We introduce SID, a versatile and scalable backpropagation-free framework that resolves update locking and reduces memory overhead through a novel local belief refinement mechanism.
2. We provide a solid theoretical foundation, proving that under well-defined conditions, SID guarantees monotonic performance improvement as network depth increases.
3. We conduct a comprehensive empirical evaluation, showing that SID matches or exceeds BP's performance, particularly showcasing enhanced performance in deeper network architectures and settings with high label noise.

## 2 RELATED WORK

We position SID relative to existing paradigms that seek to overcome the limitations of backpropagation.

**Approximating the Backward Pass.** A significant body of work has shown that the strict weight symmetry of BP is not essential for learning. Feedback Alignment (FA) and Direct Feedback Alignment (DFA) (Lillicrap et al., 2014; Nøkland, 2016) use fixed random matrices to convey error signals, while Synthetic Gradients / Decoupled Neural Interfaces (DNI) (Jaderberg et al., 2017) train local models to predict gradients from higher layers. These methods successfully break weight symmetry and can alleviate update locking. *In contrast to SID*, these methods still aim to approximate the global gradient signal. SID takes a different approach by replacing the single global objective with a set of cooperative local objectives, eliminating the need to transport or approximate gradients between modules altogether.

**Local Target-Based Learning.** Target Propagation (TP) and its variants (Lee et al., 2015) offer another approach to layer-wise training. Instead of propagating a scalar error, these methods compute layer-specific "targets" for activations using learned approximate inverses of each layer's function (e.g., auto-encoders). While effective, this approach hinges on the quality of the learned inverses, which can be unstable, especially for non-invertible or expansive layers. Our framework avoids the need for auxiliary inverse models. The "target" for each module is dynamically formed by its local objective, which smoothly interpolates between the prior belief and the ground-truth label, providing a more direct and stable learning signal.

**Alternative Training Dynamics.** Some methods fundamentally alter the network's computational process. The Forward-Forward algorithm (Hinton, 2022; Sun et al., 2025; Chen et al., 2025; Aghagolzadeh & Ezoji, 2025; Ren & Li, 2024) replaces the forward-backward dynamic with two forward passes—one for positive data, one for negative data—and updates weights based on a local goodness metric. Equilibrium Propagation (Scellier & Bengio, 2017) uses energy-based models

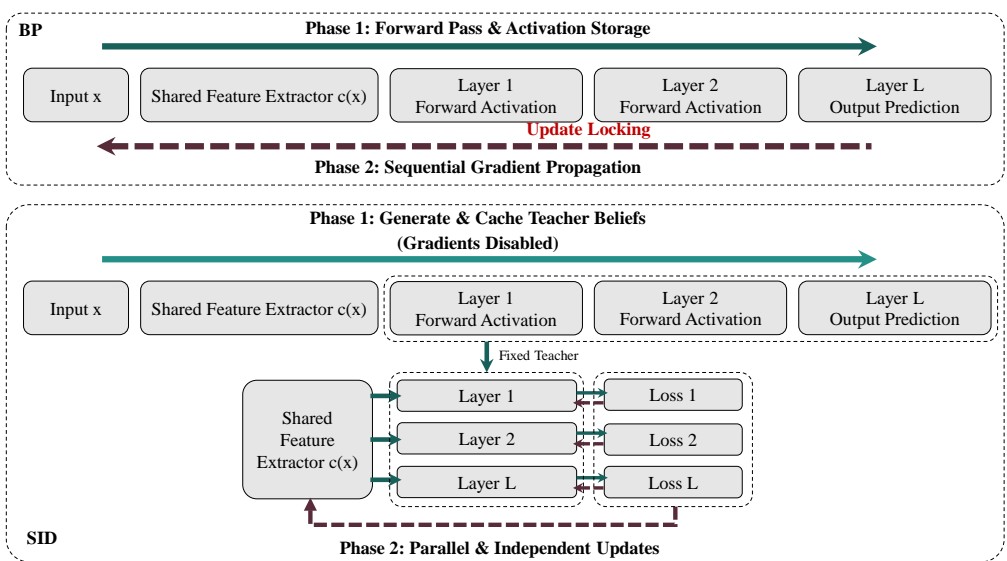

Figure 1: **Comparison of BP and SID Training Paradigms.** While BP's end-to-end gradient propagation (**top**) creates a sequential bottleneck, SID (**bottom**) introduces a two-phase process. **Phase 1** performs a single gradient-free forward pass to generate a set of fixed "teacher" beliefs. **Phase 2** then uses these local teachers to update all modules in parallel, fundamentally resolving update locking and enabling scalable training.

that converge to a fixed point, with learning signals derived from nudging the system's state. Other approaches such as PETRA (Rivaud et al., 2025) focus on parallelization and memory efficiency, while NoProp (Li et al., 2025) eliminates both forward- and back-propagation entirely. A key advantage of SID is that it preserves the standard, single-pass feed-forward architecture during inference. The modifications are confined to the training phase, making SID a non-disruptive and practical alternative for conventional deep learning applications.

**Knowledge Distillation and Layer-wise Supervision.** SID can be viewed as a structured form of *internal distillation*. The main idea of knowledge distillation is to use intermediate representations of a "teacher" to guide a "student"(Hinton et al., 2015). This idea has been extended to layer-wise self-supervision, as seen in Born-Again Networks (Furlanello et al., 2018), and more recently in hierarchical self-distillation frameworks (Gurioli et al., 2025; Xu et al., 2025; Liu et al., 2024). *SID's novel contribution* to this area is the decomposition of the distillation process into a cooperative cascade governed by both a distillation term and a local consistency term. This consistency regularizer is crucial; it ensures that learning is incremental and stable across the network's depth, a property not explicitly enforced by prior distillation or layer-wise supervision schemes.

## 3 THE SID FRAMEWORK

In this section, we formally introduce the Synergistic Information Distillation (SID) framework. We first provide a high-level overview of its architecture and the core belief refinement process. We then detail the local objective function that drives learning and conclude with the complete two-phase training algorithm.

### 3.1 FRAMEWORK OVERVIEW AND BELIEF REFINEMENT

The central idea of SID is to reframe the end-to-end learning problem as a cascade of localized cooperative learning steps. We consider a deep network composed of a shared feature extractor, denoted by $c(\cdot)$, and a pipeline of $L$ sequential processing modules, $\{f_1, \ldots, f_L\}$. The role of this pipeline is to progressively refine a probabilistic distribution over the label set $\mathcal{Y} = \{1, \ldots, m\}$, a distribution we term a "belief."

For a given input $x \in \mathcal{X}$, the process begins with an initial maximum-entropy belief, $p_0$, which is the uniform distribution over $\mathcal{Y}$. Each subsequent module $f_i$ receives the belief $p_{i-1}$ from its predecessor and the shared features $c(x)$ as input, producing a more informed belief $p_i$:

$$p_i(\cdot) = f_i\big(p_{i-1}(\cdot), c(x)\big), \quad \text{for } i = 1, \ldots, L. \tag{1}$$

This sequential refinement allows the network to incrementally build confidence. At inference time, the final belief $p_L$ is used for prediction via $\arg\max_k p_L(k)$. The key challenge, which we address next, is to design a local training signal that encourages this cooperative behavior without requiring global coordination through backpropagation.

### 3.2 THE LOCAL SID OBJECTIVE

To enable decoupled training, we define a local objective function $\mathcal{L}_i$ for each module $f_i$. This objective guides the module to produce a more accurate belief $p_i$ using only locally available information: its input belief $p_{i-1}$ and the ground-truth target distribution $p_y$. We formulate this as a weighted combination of two Kullback-Leibler (KL) divergences:

$$\mathcal{L}_i(p_{i-1}; f_i) = \underbrace{\alpha D_{\mathrm{KL}}\big(p_i \| p_y\big)}_{\text{Distillation Term}} + \underbrace{(1-\alpha) D_{\mathrm{KL}}\big(p_i \| \mathrm{sg}(p_{i-1})\big)}_{\text{Consistency Term}}. \tag{2}$$

Here, $p_i = f_i(p_{i-1}, c(x))$, $p_y$ is the one-hot distribution for the true label, $\alpha \in (0, 1)$ is a hyperparameter, and $\mathrm{sg}(\cdot)$ is the stop-gradient operator.

- **The Distillation Term** provides the primary supervisory signal, pulling the module's output belief $p_i$ towards the ground-truth target $p_y$.
- **The Consistency Term** acts as a regularizer. By penalizing large deviations from the input belief $p_{i-1}$, it encourages an incremental refinement process, preventing any single module from discarding useful information accumulated by its predecessors.

**Module Decoupling via Stop-Gradient.**  The key mechanism for decoupling modules is the **stop-gradient** operator, which we introduce and denote as $\mathrm{sg}(\cdot)$. This standard operator, available in deep learning frameworks (Paszke et al., 2019), is mathematically an identity function during the forward pass but has a zero derivative during the backward pass. That is, for any variable $\mathbf{z}$, $\mathrm{sg}(\mathbf{z}) = \mathbf{z}$ but $\nabla \mathrm{sg}(\mathbf{z}) = \mathbf{0}$.

By applying this operator to the input belief in the consistency term, we treat $\mathrm{sg}(p_{i-1})$ as a fixed constant—a non-differentiable target—during gradient computation. Consequently, when backpropagating through $\mathcal{L}_i$, the gradient flow from $p_i$ is blocked at $p_{i-1}$, severing the backward dependency to module $f_{i-1}$. This ensures that the gradient $\nabla \mathcal{L}_i$ only affects the parameters of the current module $f_i$ and the shared extractor $c$.

### 3.3 THE SID TRAINING ALGORITHM

The SID training process for a given minibatch contains two phases. We denote the parameters of the shared extractor $c$ as $\theta_c$ and the parameters of the module pipeline $\{f_i\}$ as $\boldsymbol{\theta}_{\text{modules}} = \{\theta_i\}_{i=1}^{L}$.

Training iterates over a dataset $\mathcal{D}$. For each minibatch $B = \{(x_k, y_k)\}$ sampled from $\mathcal{D}$, the process is as follows:

**Phase 1: Teacher Generation (Algorithm 1).**  The first phase generates the consistency targets for each module. Using the current network parameters, a single forward pass is executed with gradient computation disabled. This pass produces a sequence of teacher beliefs, $\mathcal{P}_{\text{teachers}} = \{P_0, P_1, \ldots, P_{L-1}\}$, which are then cached. Because gradients are disabled, these beliefs are treated as fixed non-differentiable targets in the next phase.

**Phase 2: Parallel Update (Algorithm 2).**  In the second phase, learning occurs. First, the shared features $Z$ are recomputed with gradients enabled to construct the computation graph. Then, each module $f_i$ is updated independently and in parallel. For each module $i$, its local loss $\mathcal{L}_i$ is computed using its corresponding cached teacher belief $P_{i-1}^{\text{teacher}}$ and the shared features $Z$. Gradients are then

**Algorithm 1** Generate Teacher Beliefs

1: **Input:** Minibatch $B$, parameters $\theta_c, \boldsymbol{\theta}_{\text{modules}}$.

2: **Ensure:** Gradient computation is disabled.
3: $Z_{\text{detached}} \leftarrow c(B_x; \theta_c)$
4: $P_0 \leftarrow \text{Uniform}(\mathcal{Y})$
5: $\mathcal{P}_{\text{teachers}} \leftarrow [P_0]$ {Cache list}
6: **for** $i = 1$ to $L - 1$ **do**
7: $\quad P_i \leftarrow f_i(\mathcal{P}_{\text{teachers}}[i-1], Z_{\text{detached}}; \theta_i)$
8: $\quad$ Append $P_i$ to $\mathcal{P}_{\text{teachers}}$
9: **end for**
10: **return** $\mathcal{P}_{\text{teachers}}$

**Algorithm 2** Parallel Local Updates

1: **Input:** Minibatch $B$, $\mathcal{P}_{\text{teachers}}$, params $\theta_c, \boldsymbol{\theta}_{\text{modules}}$.
2: $g_c \leftarrow 0$ {Initialize accumulator}
3: $Z \leftarrow c(B_x; \theta_c)$ {Build graph}
4: **for** $i = 1$ to $L$ **in parallel do**
5: $\quad P_{i-1}^{\text{teacher}} \leftarrow \mathcal{P}_{\text{teachers}}[i-1]$
6: $\quad P_i^{\text{pred}} \leftarrow f_i(P_{i-1}^{\text{teacher}}, Z; \theta_i)$
7: $\quad P_y \leftarrow \text{OneHot}(B_y)$
8: $\quad \mathcal{L}_i \leftarrow \alpha D_{\text{KL}}(P_i^{\text{pred}} \| P_y) + (1 - \alpha) D_{\text{KL}}(P_i^{\text{pred}} \| P_{i-1}^{\text{teacher}})$
9: $\quad g_i \leftarrow \nabla_{\theta_i} \mathcal{L}_i$
10: $\quad g_c \leftarrow g_c + \nabla_{\theta_c} \mathcal{L}_i$ {Accumulate}
11: **end for**
12: **return** $\{g_i\}_{i=1}^L, g_c$

calculated: $\nabla_{\theta_i} \mathcal{L}_i$ for the module's own parameters and $\nabla_{\theta_c} \mathcal{L}_i$ for the shared extractor's parameters. The gradients from all modules with respect to the shared extractor are accumulated into $g_c$.

Finally, after all local gradients are computed, the parameters $\theta_c$ and all $\{\theta_i\}_{i=1}^L$ are updated in a single optimization step using the collected gradients $\{g_i\}$ and $g_c$. This two-phase design effectively eliminates the sequential dependencies inherent in BP, directly addressing update locking and memory consumption bottlenecks.

## 4 THEORETICAL ANALYSIS

We now provide a theoretical analysis of the SID framework to formally establish its core properties. Our goal is to demonstrate why SID serves as a robust and scalable alternative to backpropagation. We structure our analysis into three parts: convergence properties in an ideal setting, stability guarantees under practical conditions, and a formal complexity analysis.

To facilitate the analysis, we define the local objective as a functional $\mathcal{S}_i$. A functional is a function that takes another function—in this case, a probability distribution $p$—as its input and returns a scalar value. For a given prior belief $p_{i-1}$ and target distribution $p_y$, the functional is:

$$\mathcal{S}_i(p) \triangleq \alpha D_{\text{KL}}(p \| p_y) + (1 - \alpha) D_{\text{KL}}(p \| p_{i-1}). \tag{3}$$

The term $\mathcal{S}_i(p)$ represents the value of the local loss for any module that outputs a belief distribution $p$. The training goal for module $f_i$ is thus to find parameters $\theta_i$ that produce an output $p_i$ that minimizes this functional.

### 4.1 CONVERGENCE AND OPTIMALITY ANALYSIS

We begin by characterizing the behavior of an ideal SID pipeline, which we define as a pipeline where every module perfectly minimizes its local objective. This idealized scenario allows us to understand the fundamental convergence properties of the cooperative refinement process, abstracting away from optimization imperfections.

**Assumption 1** (Optimal Local Updates). *We assume each module $f_i$ is sufficiently expressive and perfectly optimized such that its output $p_i$ is the unique minimizer of its local objective functional $\mathcal{S}_i(p)$, i.e., $p_i = \arg\min_p \mathcal{S}_i(p)$. This assumption will be relaxed in Section 4.2.*

Under this assumption, the cascade of local refinements leads to exponential convergence towards the target distribution.

**Proposition 1** (Closed-Form Cascade and Exponential Convergence). *Suppose Assumption 1 holds. Given an initial uniform belief $p_0$, the belief $p_i$ after module $i$ is a geometric interpolation between*

$p_0$ and $p_y$:

$$p_i(k) \propto p_0(k)^{(1-\alpha)^i} p_y(k)^{1-(1-\alpha)^i}, \quad \forall k \in \mathcal{Y}. \tag{4}$$

*The symbol $\propto$ denotes proportionality up to a normalization constant. Consequently, as the number of modules $i \to \infty$, the belief $p_i$ converges pointwise to the target distribution $p_y$. The rate of convergence is geometric, determined by the factor $(1-\alpha)$.*

*Proof.* See Appendix A.1. □

**Interpretation.** Proposition 1 provides the theoretical foundation for SID's effectiveness. It demonstrates that a sequence of local greedy optimizations can collectively solve the global learning task. Even though each module performs only a small conservative update governed by $\alpha$, their composition rapidly concentrates the belief mass onto the correct label. While this optimality may not hold in practice, the result establishes that the underlying mechanism of SID is sound.

## 4.2 Robustness and Stability Guarantees

In practice, neural network modules trained with finite data and gradient-based methods will not perfectly minimize their local objectives. A crucial property for any practical algorithm is robustness to such imperfections. We now show that SID maintains a strong stability guarantee under a much weaker condition: that each module simply *improves* upon its local objective.

**Proposition 2** (Monotonic Descent Guarantee). *Suppose that a module's output $p_i$ satisfies the local improvement condition $\mathcal{S}_i(p_i) \leq \mathcal{S}_i(p_{i-1})$. Then, the KL divergence to the target is guaranteed to be non-increasing. Summing this guarantee over all $L$ modules yields a telescoping bound for the entire network:*

$$D_{\mathrm{KL}}(p_L \| p_y) \leq D_{\mathrm{KL}}(p_0 \| p_y) - \frac{1-\alpha}{\alpha} \sum_{i=1}^{L} D_{\mathrm{KL}}(p_i \| p_{i-1}). \tag{5}$$

*Proof.* See Appendix A.2. □

**Interpretation.** This result is central to SID's practicality. It guarantees that as long as each local update is productive (i.e., reduces the local loss, a condition easily met with a sufficiently small learning rate), the overall network's performance with respect to the true target will not degrade with added depth. This ensures a stable and well-behaved training process, preventing catastrophic divergence. Furthermore, for the network to converge, the term $\sum_i D_{\mathrm{KL}}(p_i \| p_{i-1})$ must be bounded, which implies that $D_{\mathrm{KL}}(p_i \| p_{i-1})$ must approach zero for deep networks. This indicates that the belief sequence becomes asymptotically stationary, ensuring a stable flow of information.

## 4.3 Scalability and Parallelism Analysis

Finally, we analyze SID's computational complexity to demonstrate how it addresses the scalability bottlenecks of backpropagation. We consider a network of $L$ modules executed on a system with $P$ parallel processing devices.

**Proposition 3** (Computational and Memory Complexity). *Let $C_f^{(i)}$ and $C_b^{(i)}$ be the forward and backward computation costs for module $i$, and let $A_i$ be its activation memory cost.*

1. ***Time Complexity (Speedup):*** *In a parallel setting ($P \geq L$), the training time for SID is $T_{\mathrm{SID}} \approx \sum_{i=1}^{L} C_f^{(i)} + \max_{i=1}^{L} C_b^{(i)}$, whereas for BP it is $T_{\mathrm{BP}} = \sum_{i=1}^{L}(C_f^{(i)} + C_b^{(i)})$. SID offers a theoretical speedup by parallelizing the backward passes, eliminating the sequential dependency that causes **update locking**.*
2. ***Memory Complexity (Savings):*** *The peak activation memory required by BP is $M_{\mathrm{BP}} \approx \sum_{i=1}^{L} A_i$. With SID on $P \geq L$ devices, the per-device peak memory is $M_{\mathrm{SID}} \approx \max_{i=1}^{L} A_i$. For deep networks ($L \gg 1$), this constitutes a substantial reduction in **memory consumption**, scaling with the size of the largest module rather than the full network depth.*

*Proof.* See Appendix A.3. □

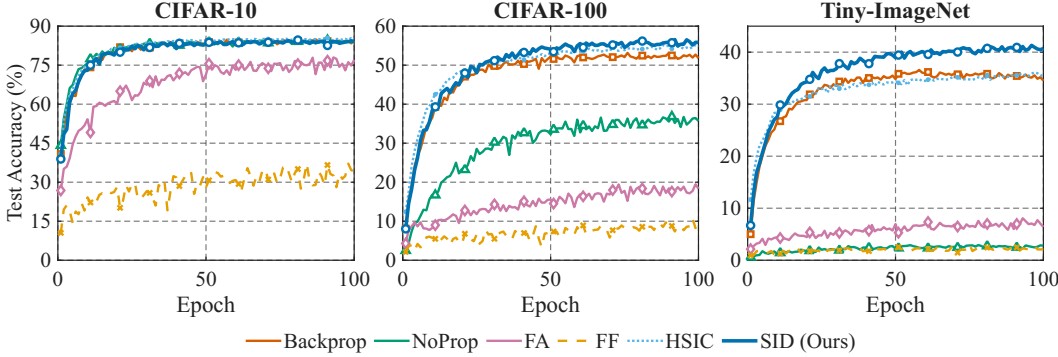

Figure 2: Test accuracy convergence curves on CIFAR-10 (left), CIFAR-100 (center), and Tiny-ImageNet (right). SID (blue, bold) demonstrates superior performance on more complex datasets. This performance gain is achieved alongside SID's significant reductions in time and memory complexity (see Section 4.3).

**Interpretation.** Proposition 3 directly connects SID's design to its core motivations. By breaking the end-to-end gradient chain, SID's architecture allows the computationally intensive backward pass to be parallelized. This architectural advantage, combined with the dramatically reduced memory footprint, makes SID an inherently more scalable training framework than standard backpropagation, particularly for very deep or large models.

## 5 EXPERIMENTS

We conduct a comprehensive empirical evaluation to validate the performance, scalability, and robustness of SID.

### 5.1 EXPERIMENTAL SETUP

We evaluate SID against standard **BP** and a suite of backpropagation-free baselines (**NoProp** (Li et al., 2025), **FA** (Nøkland, 2016), **FF** (Hinton, 2022), **HSIC** (Ma et al., 2019)) on CIFAR-10, CIFAR-100 (Krizhevsky, 2009), and Tiny-ImageNet (Le & Yang, 2015). To create a challenging optimization benchmark, we designed a VGG-style SimpleCNN composed of a shared convolutional extractor and a deep stack of MLP modules. Its deep structure without residual connections makes it susceptible to optimization difficulties, thereby allowing for a rigorous evaluation of each training method's effectiveness. For generality, we also report the results on standard architectures. All experiments are averaged over three random seeds. (See Appendix B.1.2 for detailed architecture and setup descriptions).

### 5.2 PERFORMANCE AND SCALABILITY ANALYSIS

**Observation 1: The performance improvement of SID grows with task complexity.** As shown in Table 1, SID's performance scales effectively with task difficulty. On CIFAR-10, SID's accuracy is statistically on par with the BP and strong HSIC baselines. However, as task complexity increases, SID establishes a clear advantage, outperforming BP by 4.3% on CIFAR-100 and 6.8% on Tiny-ImageNet. The convergence curves in Figure 2 further illustrate this: on more complex datasets, SID's accuracy consistently improves while BP's learning curve begins to plateau. This suggests SID's local cooperative learning mechanism is more effective at navigating complex loss landscapes.

**Observation 2: SID resolves update locking, enabling strong parallel scalability.** A core motivation for SID is to eliminate the sequential dependency of the backward pass. To quantify this benefit, we conducted a computational profiling experiment. We first timed the forward and backward passes of each component on a single GPU to isolate pure computational costs from communication overhead. We then projected these timings to a multi-GPU scenario assuming ideal model parallelism, where modules are distributed across processors(Chen et al.,

Table 1: Final test accuracy (%) on vision benchmarks, reported as mean ± std over three seeds. The best-performing local method is in bold, second-best is underlined. Performance improvement of SID over BP is shown in blue.

| Method | CIFAR-10 | CIFAR-100 | Tiny-ImageNet |
|---|---|---|---|
| *Global Gradient Methods* | | | |
| Backpropagation (BP) | $84.32 \pm 0.45$ | $51.81 \pm 0.52$ | $34.51 \pm 0.61$ |
| *Local Gradient Methods* | | | |
| Feedback Alignment (FA) | $77.03 \pm 0.68$ | $18.32 \pm 0.75$ | $6.39 \pm 0.42$ |
| **SID (Ours)** | $\underline{84.42 \pm 0.35}$ | $\mathbf{56.12 \pm 0.38}$ (**+4.31**) | $\mathbf{41.37 \pm 0.45}$ (**+6.86**) |
| *Purely Local / Forward-Only Methods* | | | |
| NoProp | $84.18 \pm 0.33$ | $35.69 \pm 0.59$ | $2.80 \pm 0.25$ |
| Forward-Forward (FF) | $34.10 \pm 0.81$ | $9.13 \pm 0.48$ | $2.13 \pm 0.19$ |
| HSIC-based | $\mathbf{85.21 \pm 0.29}$ | $\underline{55.18 \pm 0.41}$ | $\underline{35.61 \pm 0.38}$ |

2016; Rajbhandari et al., 2020; Goyal et al., 2018; Dean et al., 2012) (see Appendix B.3). This analysis yields a *theoretical speedup* which quantifies the architectural advantage of SID's parallelizable design. As shown in Figure 3, the speedup scales robustly with the number of processors ($P$) and network depth ($L$). For a deep network with $L = 64$, the projected speedup approaches 2.4x, confirming that SID's local updates effectively address the BP bottleneck.

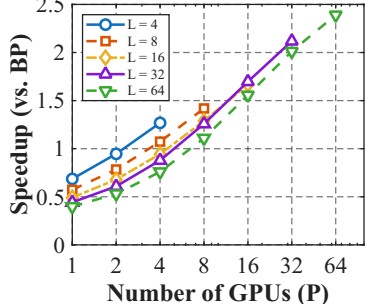

**Observation 3: SID demonstrates superior stability with network depth.** SID's architectural advantages also lead to improved optimization stability in deep networks. To evaluate this, we trained the SimpleCNN on CIFAR-100 at varying depths. This architecture, lacking residual connections, creates a difficult optimization landscape that highlights potential training failures. As shown in Figure 4 (left), SID's accuracy improves monotonically with network depth. In contrast, BP's performance degrades after 8 modules, a classic sign of optimization failure in very deep networks. On a ResNet backbone (right), where skip-connections mitigate this issue for BP, SID still shows more consistent improvement, further highlighting the inherent stability of its local regularized learning process.

Figure 3: Theoretical speedup of SID over BP.

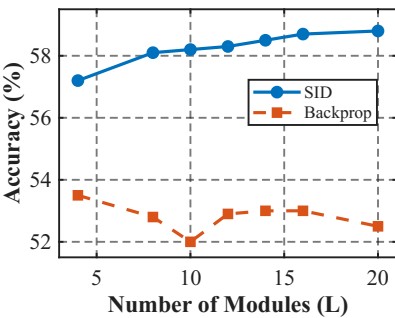
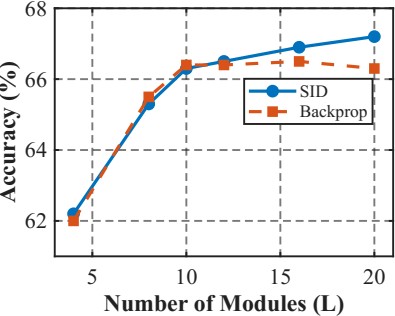

Figure 4: Depth scaling on CIFAR-100. **Left:** On a SimpleCNN, SID's accuracy improves with depth while BP's degrades. **Right:** On a ResNet, SID shows more consistent improvement.

**Generality and Ablations.** SID is a general-purpose strategy that exhibits a consistent performance advantage over BP across modern architectures, including VGG (Simonyan & Zisserman, 2015), ResNet (He et al., 2015), and ViT (Dosovitskiy et al., 2021) (see Table 3 in Ap-

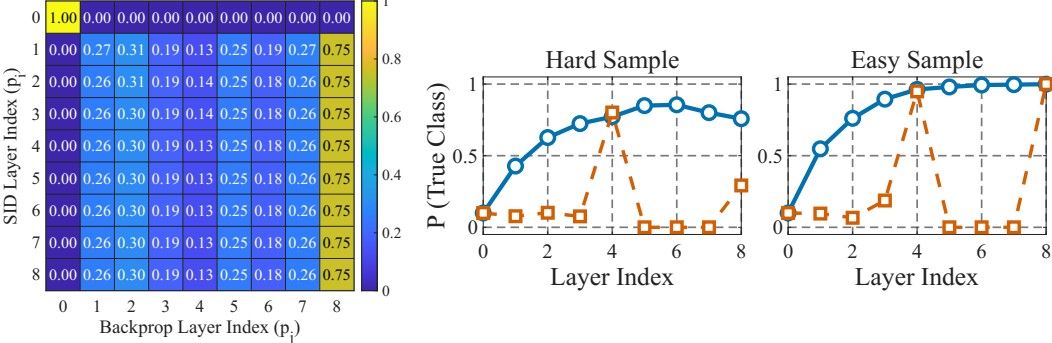

Figure 5: **Left:** CKA similarity between SID and BP beliefs. The vertical stripe in the final column highlights SID's "converge-then-refine" dynamic. **Right:** Belief evolution on hard examples (misclassified by at least one model). SID's belief in the true class (blue) shows a more stable and monotonic progression than BP's (orange).

pendix B.2.1).Ablation studies confirm the robustness of the hyperparameter $\alpha$ and the necessity of cooperatively updating the shared extractor with gradients from all modules (see Appendix B.2.2).

## 5.3 ANALYSIS OF INTERNAL MECHANISMS

**SID exhibits a "converge-then-refine" learning dynamic.** To understand the performance differences between SID and BP, we analyzed their internal layer-wise belief representations using Centered Kernel Alignment (CKA)(Kornblith et al., 2019). For this analysis, we regard the softmax output of any given layer as its probabilistic "belief." The CKA heatmap in Figure 5 (left) reveals a distinct structural difference in how representations are formed. SID's early layers (e.g., layers 1-3, y-axis) already exhibit high similarity to BP's *final* layer belief ($p_8$, x-axis), evidenced by the strong vertical stripe in the last column. This indicates that SID's network produces a high-quality prediction candidate early in its hierarchy, which subsequent layers then incrementally refine.

This phenomenon is further explored in Figure 5 (right), which tracks belief evolution for "hard" samples (those misclassified by at least one method). On these samples, SID's belief in the true class often increases monotonically and smoothly. Conversely, BP's belief can be erratic, with sharp drops in intermediate layers (similar observations exist in representation dynamics analyses). This stable, incremental refinement, enforced by the local consistency term in SID's objective, likely contributes to its enhanced robustness and superior performance on challenging tasks.

## 6 CONCLUSION

In this work, we introduced SID, a novel training framework that resolves the critical update locking and memory consumption bottlenecks of BP by reframing learning as a cooperative cascade of local belief refinements. By decoupling modules via a stop-gradient on a local consistency objective, SID enables memory-efficient and parallelizable training without altering the standard inference pass. Our theoretical analysis demonstrates a monotonic descent guarantee of SID, where this guarantee ensures robust and stable training. Empirically, we demonstrated that SID's performance matches or surpasses that of BP, with its advantage growing significantly on more complex tasks and deeper networks, all while having lower computational complexity. Further analysis revealed SID's unique "converge-then-refine" learning dynamic, confirming its distinctness from end-to-end training. While this work establishes a strong foundation, limitations exist. Our empirical validation is primarily focused on image classification, and the full extent of SID's applicability to other domains, such as natural language processing or reinforcement learning, remains to be explored. Future research should therefore focus on adapting and evaluating SID on these diverse tasks and architectures. A particularly promising direction will be to leverage SID's inherent scalability to train large-scale foundation models, where its benefits could be most transformative.In summary, our results establish SID as a powerful, scalable, and practical alternative to BP, offering promising techniques for training large-scale models.

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

## LLM USAGE STATEMENT

In preparing this paper, Large Language Models (LLMs) were used as an assistive tool for grammar refinement and clarity improvements in writing. The scientific contributions, research design, experiments, and analysis were entirely conducted by the authors. LLMs did not contribute to research ideation, algorithm design, or empirical results. All content has been reviewed and validated by the authors, who take full responsibility for its accuracy.

## A  PROOFS FOR THEORETICAL ANALYSIS

In this appendix, we provide the detailed proofs for the propositions presented in Section 4. Throughout the proofs, we denote a discrete probability distribution over $\mathcal{Y} = \{1, \ldots, m\}$ as a vector $p \in \Delta^{m-1}$, where $\Delta^{m-1}$ is the $(m-1)$-simplex. All KL divergences assume the distributions have full support, which can be ensured in practice with label smoothing.

### A.1  PROOF OF PROPOSITION 1

Proposition 1 relies on first finding the closed-form minimizer of the local objective $\mathcal{S}_i(p)$. Let's call this intermediate result Lemma 1.

**Lemma 1** (Closed-Form Local Minimizer). *Given fixed distributions $p_{i-1}$ and $p_y$ with full support, the functional $\mathcal{S}_i(p) = \alpha D_{\mathrm{KL}}(p\|p_y) + (1-\alpha)D_{\mathrm{KL}}(p\|p_{i-1})$ is strictly convex. Its unique minimizer, $p_i^\star \triangleq \arg\min_p \mathcal{S}_i(p)$, is given by the normalized power mean (geometric interpolation):*

$$p_i^\star(k) = \frac{1}{Z_i}p_{i-1}(k)^{1-\alpha}p_y(k)^\alpha, \quad \text{where } Z_i = \sum_{j=1}^{m} p_{i-1}(j)^{1-\alpha}p_y(j)^\alpha. \tag{6}$$

*Equivalently, this can be expressed as $p_i^\star(k) \propto p_{i-1}(k)^{1-\alpha}p_y(k)^\alpha$ for each $k \in \mathcal{Y}$.*

*Proof of Lemma 1.* The KL divergence is strictly convex, and a non-negative weighted sum of strictly convex functions is also strictly convex. Thus, $\mathcal{S}_i(p)$ is strictly convex and has a unique minimizer. To find it, we can use Lagrange multipliers to enforce the constraint $\sum_k p(k) = 1$. The Lagrangian is:

$$\mathcal{L}(p, \lambda) = \sum_k p(k) \log \frac{p(k)^\alpha}{p_y(k)^\alpha} + \sum_k p(k) \log \frac{p(k)^{1-\alpha}}{p_{i-1}(k)^{1-\alpha}} + \lambda\left(1 - \sum_k p(k)\right).$$

Taking the derivative with respect to $p(k)$ and setting it to zero:

$$\frac{\partial \mathcal{L}}{\partial p(k)} = \alpha(\log p(k) + 1) - \alpha \log p_y(k) + (1-\alpha)(\log p(k) + 1) - (1-\alpha)\log p_{i-1}(k) - \lambda = 0$$

$$\implies \log p(k) + 1 - \alpha \log p_y(k) - (1-\alpha)\log p_{i-1}(k) = \lambda$$

$$\implies \log p(k) = \lambda - 1 + \log\left(p_{i-1}(k)^{1-\alpha}p_y(k)^\alpha\right)$$

$$\implies p(k) = \exp(\lambda - 1) \cdot p_{i-1}(k)^{1-\alpha}p_y(k)^\alpha.$$

Let $1/Z_i = \exp(\lambda - 1)$. This constant is determined by the summation constraint $\sum_k p(k) = 1$, which gives $Z_i = \sum_j p_{i-1}(j)^{1-\alpha}p_y(j)^\alpha$. This completes the proof of the lemma. $\square$

*Proof of Proposition 1.* We proceed by induction. From Lemma 1, under Assumption 1, the output of module $i$ is $p_i(k) \propto p_{i-1}(k)^{1-\alpha}p_y(k)^\alpha$.

**Base case (i=1):**
$$p_1(k) \propto p_0(k)^{1-\alpha}p_y(k)^\alpha.$$

This matches the formula $p_i(k) \propto p_0(k)^{(1-\alpha)^i}p_y(k)^{1-(1-\alpha)^i}$ for $i = 1$, since $1 - (1-\alpha)^1 = \alpha$.

**Inductive step:** Assume the formula holds for $i - 1$: $p_{i-1}(k) \propto p_0(k)^{(1-\alpha)^{i-1}} p_y(k)^{1-(1-\alpha)^{i-1}}$. Then for module $i$:

$$
\begin{aligned}
p_i(k) &\propto p_{i-1}(k)^{1-\alpha} p_y(k)^{\alpha} \\
&\propto \left( p_0(k)^{(1-\alpha)^{i-1}} p_y(k)^{1-(1-\alpha)^{i-1}} \right)^{1-\alpha} p_y(k)^{\alpha} \\
&\propto p_0(k)^{(1-\alpha)^{i-1}(1-\alpha)} p_y(k)^{(1-(1-\alpha)^{i-1})(1-\alpha)} p_y(k)^{\alpha} \\
&\propto p_0(k)^{(1-\alpha)^i} p_y(k)^{1-\alpha-(1-\alpha)^i+\alpha} \\
&\propto p_0(k)^{(1-\alpha)^i} p_y(k)^{1-(1-\alpha)^i}.
\end{aligned}
$$

This completes the induction. For convergence, since $\alpha \in (0,1)$, we have $(1-\alpha) \in (0,1)$. Thus, as $i \to \infty$, the exponent $(1-\alpha)^i \to 0$. The exponent of $p_y(k)$, which is $1 - (1-\alpha)^i$, goes to 1. Therefore, $p_i(k) \propto p_0(k)^0 p_y(k)^1 = p_y(k)$. Since this holds for all $k$, $p_i$ converges to $p_y$. The convergence rate is determined by how quickly $(1-\alpha)^i$ approaches zero, which is geometric. $\square$

### A.2 PROOF OF PROPOSITION 2

We are given the local improvement condition: $\mathcal{S}_i(p_i) \leq \mathcal{S}_i(p_{i-1})$. Let's expand both sides of the inequality:

$$
\begin{aligned}
\mathcal{S}_i(p_i) &= \alpha D_{\mathrm{KL}}(p_i \| p_y) + (1-\alpha) D_{\mathrm{KL}}(p_i \| p_{i-1}) \\
\mathcal{S}_i(p_{i-1}) &= \alpha D_{\mathrm{KL}}(p_{i-1} \| p_y) + (1-\alpha) D_{\mathrm{KL}}(p_{i-1} \| p_{i-1}) = \alpha D_{\mathrm{KL}}(p_{i-1} \| p_y).
\end{aligned}
$$

Substituting these into the condition gives:

$$
\alpha D_{\mathrm{KL}}(p_i \| p_y) + (1-\alpha) D_{\mathrm{KL}}(p_i \| p_{i-1}) \leq \alpha D_{\mathrm{KL}}(p_{i-1} \| p_y).
$$

Since $D_{\mathrm{KL}}(p_i \| p_{i-1}) \geq 0$ and $\alpha \in (0,1)$, we can rearrange the inequality to isolate $D_{\mathrm{KL}}(p_i \| p_y)$:

$$
\alpha D_{\mathrm{KL}}(p_i \| p_y) \leq \alpha D_{\mathrm{KL}}(p_{i-1} \| p_y) - (1-\alpha) D_{\mathrm{KL}}(p_i \| p_{i-1}).
$$

Dividing by $\alpha > 0$ yields the single-step descent inequality:

$$
D_{\mathrm{KL}}(p_i \| p_y) \leq D_{\mathrm{KL}}(p_{i-1} \| p_y) - \frac{1-\alpha}{\alpha} D_{\mathrm{KL}}(p_i \| p_{i-1}).
$$

To obtain the telescoping bound for the entire network, we sum this inequality from $i = 1$ to $L$:

$$
\sum_{i=1}^{L} \left( D_{\mathrm{KL}}(p_i \| p_y) - D_{\mathrm{KL}}(p_{i-1} \| p_y) \right) \leq -\frac{1-\alpha}{\alpha} \sum_{i=1}^{L} D_{\mathrm{KL}}(p_i \| p_{i-1}).
$$

$$
\left( D_{\mathrm{KL}}(p_L \| p_y) - D_{\mathrm{KL}}(p_{L-1} \| p_y) \right) + \cdots + \left( D_{\mathrm{KL}}(p_1 \| p_y) - D_{\mathrm{KL}}(p_0 \| p_y) \right) \leq -\frac{1-\alpha}{\alpha} \sum_{i=1}^{L} D_{\mathrm{KL}}(p_i \| p_{i-1}).
$$

The sum on the left is a telescoping series, which simplifies to $D_{\mathrm{KL}}(p_L \| p_y) - D_{\mathrm{KL}}(p_0 \| p_y)$.

$$
D_{\mathrm{KL}}(p_L \| p_y) - D_{\mathrm{KL}}(p_0 \| p_y) \leq -\frac{1-\alpha}{\alpha} \sum_{i=1}^{L} D_{\mathrm{KL}}(p_i \| p_{i-1}).
$$

Rearranging gives the final bound:

$$
D_{\mathrm{KL}}(p_L \| p_y) \leq D_{\mathrm{KL}}(p_0 \| p_y) - \frac{1-\alpha}{\alpha} \sum_{i=1}^{L} D_{\mathrm{KL}}(p_i \| p_{i-1}).
$$

This completes the proof.

### A.3 PROOF SKETCH FOR PROPOSITION 3

This proposition follows directly from the definition of the SID and BP training algorithms.

**Time Complexity.** The total time for one BP minibatch is the sum of all forward and backward passes, as they must be executed sequentially: $T_{\text{BP}} = \sum_{i=1}^{L}(C_f^{(i)} + C_b^{(i)})$. For SID, the teacher forward pass is sequential: $T_{\text{fwd}} = \sum_{i=1}^{L} C_f^{(i)}$. The local updates, however, can run in parallel. On a system with $P \geq L$ devices, all $L$ backward passes can execute concurrently. The time for this phase is limited by the slowest module: $T_{\text{bwd}} = \max_i C_b^{(i)}$. The total time is their sum (ignoring communication for aggregation, which is common to both), $T_{\text{SID}} \approx T_{\text{fwd}} + T_{\text{bwd}}$. The speedup comes from replacing the sum of backward costs $\sum C_b^{(i)}$ with the max-cost $\max_i C_b^{(i)}$.

**Memory Complexity.** For BP, to compute the gradient for the first module, all activations from all subsequent modules $(A_1, \ldots, A_L)$ must be kept in memory. The peak memory is therefore the sum of all activation memory costs, $M_{\text{BP}} \approx \sum_{i=1}^{L} A_i$. For SID, when computing the local gradient for module $f_i$ on device $i$, only its own activations $A_i$ need to be stored. The gradient does not propagate to other modules, so their activations can be discarded. Therefore, the peak memory requirement for any single device is simply the memory needed for the largest module it hosts, $\max_i A_i$.

# B   DETAILED EXPERIMENTAL INFORMATION

This appendix provides comprehensive details regarding the experimental setup, supplemental results, and analysis methodologies discussed in the main paper. Our goal is to ensure full reproducibility and provide deeper insights into our findings.

## B.1   EXPERIMENTAL SETUP DETAILS

### B.1.1   DATASETS AND PREPROCESSING

We used three standard image classification benchmarks. Standard data augmentation techniques were applied during training for all datasets.

- **CIFAR-10 & CIFAR-100:** For both datasets (Krizhevsky, 2009), we used the standard training set of 50,000 images and a test set of 10,000 images. The images are 32x32 pixels. For data augmentation, we applied random horizontal flips and random 32x32 crops from images zero-padded to 40x40.
- **Tiny-ImageNet:** This dataset (Le & Yang, 2015) contains 200 classes from ImageNet, with 100,000 training images, 10,000 validation images, and 10,000 test images. Images are down-scaled to 64x64 resolution. For augmentation, we used random horizontal flips and random 64x64 crops from images zero-padded to 72x72.

All images were normalized using the per-channel mean and standard deviation computed from their respective training sets.

### B.1.2   ARCHITECTURES

**SimpleCNN Architecture.** Our primary testbed, the SimpleCNN, was designed with a deep stack of processing modules to rigorously test optimization algorithms. It separates perceptual feature extraction from sequential belief refinement. The architecture is detailed in Table 2.

**Standard Architectures.** For the architectural generality experiments, we used standard, off-the-shelf implementations of VGG-11, ResNet-18, and a ViT-Tiny model adapted for CIFAR-scale images.

### B.1.3   BASELINE IMPLEMENTATION DETAILS

- **Backpropagation (BP):** Standard end-to-end training of the entire network with a global cross-entropy loss.
- **Feedback Alignment (FA):** Implemented by replacing the transpose of the weight matrices in the backward pass with fixed random matrices, which are initialized once and remain frozen.

Table 2: The SimpleCNN architecture. The network consists of a shared feature extractor followed by $L$ identical processing modules.

| Component | Layer Type | Output Shape / Dims |
|---|---|---|
| **Shared Feature Extractor ($c$)** | | |
| | Conv(3, 32, k=3), ReLU | (B, 32, H, W) |
| | Conv(32, 64, k=3), ReLU, MaxPool(2) | (B, 64, H/2, W/2) |
| | Conv(64, 128, k=3), ReLU | (B, 128, H/2, W/2) |
| | Conv(128, 128, k=3), ReLU, MaxPool(2) | (B, 128, H/4, W/4) |
| | AdaptiveAvgPool(1), Flatten | (B, 128) |
| | Linear(128, 128) | (B, 128) $\rightarrow z$ |
| **Processing Module ($f_i$), repeated L times** | | |
| | Input Concatenation | (B, 128 + num_classes) -¿ cat($p_{i-1}, z$) |
| | Linear, ReLU | (B, 256) |
| | Linear | (B, num_classes) $\rightarrow$ Logits for $p_i$ |

- **NoProp:** A state-of-the-art local learning method where each module is trained independently to denoise a noisy version of the final target, conditioned on the input image, thereby eliminating the need for any inter-module signal propagation during training.

- **Forward-Forward (FF):** We implemented a version where each module is a layer trained to have a higher sum-of-squares activity for "positive" data (correct label overlaid) than for "negative" data (incorrect label overlaid), using a margin-based loss.

- **HSIC-based Learning:** Each module is trained to maximize the Hilbert-Schmidt Independence Criterion between its output features and the target labels, using an unbiased estimator with an RBF kernel.

### B.1.4 TRAINING PROTOCOL

To ensure fair comparisons, all methods were trained using the same optimization protocol unless otherwise specified.

- **Optimizer:** Adam (Kingma & Ba, 2017).

- **Learning Rate:** An initial learning rate of $1 \times 10^{-3}$.

- **Learning Rate Schedule:** A cosine annealing schedule over the course of training.

- **Batch Size:** 128 for all datasets.

- **Epochs:** 100 epochs for benchmark comparisons; varied for scalability studies.

- **Loss Function:** Cross-entropy with label smoothing ($\epsilon = 0.1$) for all applicable methods.

- **SID Hyperparameter $\alpha$:** Based on a validation sweep (see Figure 6), we used a fixed value of $\alpha = 0.5$ for all main experiments.

### B.1.5 COMPUTING INFRASTRUCTURE

All experiments were conducted on a server equipped with NVIDIA A100 GPUs using PyTorch.

### B.2 SUPPLEMENTAL EXPERIMENTAL RESULTS

### B.2.1 ARCHITECTURAL GENERALITY

Table 3 provides the full results for the architectural generality experiment, demonstrating that SID's competitive performance is not limited to the SimpleCNN architecture but extends to modern standard models.

Table 3: Full results for Architectural Generality. Accuracy (%) of SID vs. BP on standard backbones, reported as mean ± std over three seeds.

| Dataset | Backbone | Backpropagation (BP) | SID (Ours) |
|---|---|---|---|
| CIFAR-10 | SimpleCNN | 84.31 ± 0.90 | 84.37 ± 0.26 |
| | VGG-11 | 88.21 ± 0.15 | **88.26 ± 0.18** |
| | ResNet-18 | 91.12 ± 0.21 | **91.15 ± 0.25** |
| | ViT-Tiny | 92.35 ± 0.11 | **93.11 ± 0.19** |
| CIFAR-100 | SimpleCNN | 52.00 ± 0.65 | **54.05 ± 0.11** |
| | VGG-11 | 61.01 ± 0.24 | **62.33 ± 0.31** |
| | ResNet-18 | 66.24 ± 0.28 | **67.37 ± 0.33** |

### B.2.2 ABLATION STUDIES

Figure 6 presents the detailed ablation studies for the hyperparameter $\alpha$ and the feature extractor update strategy. The left panel shows that SID is robust to the choice of $\alpha$, achieving high performance for a wide range of values between 0.2 and 0.8. The right panel demonstrates that the default "all-layer" update strategy for the shared extractor is critical for optimal performance, significantly outperforming variants where the extractor is frozen or updated only by the final layer's loss. This confirms that allowing all modules to cooperatively refine the shared features is a key component of SID's success.

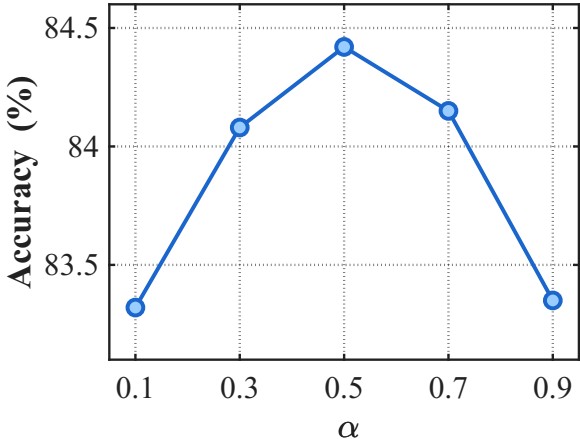

Figure 6: Impact of the consistency parameter $\alpha$ on CIFAR-10 accuracy. Performance is robust, peaking around $\alpha = 0.5$.

| Update Strategy | CIFAR-10 (%) | CIFAR-100 (%) |
|---|---|---|
| SID (default) | **84.42** | **54.05** |
| SID (final_layer) | 83.85 | 52.15 |
| SID (frozen) | 82.90 | 49.80 |

Table 4: Ablation on the feature extractor update strategy. Updating with gradients from all modules (default) is critical for performance.

### B.2.3 ADDITIONAL VISUALIZATIONS FOR INTERNAL MECHANISMS

**Shared Feature Space Visualization.** Figure 7 provides a t-SNE visualization of the 128-dimensional shared feature space learned by the extractor $c(x)$ for both SID and BP on the CIFAR-10 test set. In both cases, the extractor learns a high-quality embedding where classes form largely separable clusters. This confirms that the aggregated gradient signal from all local modules in SID is

sufficient to train a powerful and discriminative shared representation, comparable to that learned with a single global loss signal in BP.

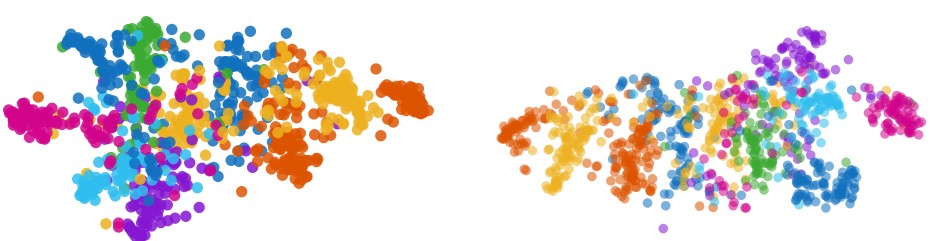

**SID Feature Space** $c(x)$      **Backprop Feature Space** $c(x)$

Figure 7: t-SNE visualization of the shared feature space $c(x)$ learned by SID (left) and BP (right). Both methods learn a well-structured embedding.

**Detailed Belief Evolution Trajectories.** Figure 8 provides additional examples of the layer-wise belief evolution for the true class, supplementing the discussion in the main paper. The plots consistently show that on challenging examples (where BP makes an incorrect final prediction, or its confidence dips significantly), SID's belief trajectory (blue solid line) is more stable and monotonically non-decreasing. This illustrates the regularizing effect of the consistency term, which prevents drastic changes in belief and encourages a more "deliberative" refinement process.

### B.3 COMPUTATIONAL PERFORMANCE ANALYSIS METHODOLOGY

This section details the methodology used for the computational profiling experiment designed to quantify the impact of update locking (Figure 3 in the main text).

**Profiling on a Single GPU.** Since direct measurement on a multi-GPU system can be confounded by system-specific communication overheads, we opted for a more controlled approach by profiling the computational components on a single GPU. This allows us to isolate the algorithmic structure as the primary variable. We used 'torch.cuda.Event' for high-precision timing of GPU operations, which avoids synchronization issues.

**Decomposition of Training Steps.** We decomposed a single minibatch update into its fundamental computational stages for both algorithms:

- **For Backpropagation (BP)**, the process is inherently sequential. We measured:
    1. $T_{BP\_fwd}$: The time for the complete forward pass, from input to loss calculation.
    2. $T_{BP\_bwd}$: The time for the complete backward pass ('loss.backward()').
- **For SID**, the process has sequential and parallelizable stages. We measured:
    1. $T_{SID\_fwd\_teacher}$: The time for the sequential gradient-free teacher forward pass.
    2. $T_{SID\_fwd\_grad}$: The time to recompute the shared features with gradients enabled.
    3. $\{T_{SID\_bwd\_i}\}_{i=1}^{L}$: The individual backward pass time for *each* of the $L$ modules, measured in isolation.

**Theoretical Performance Projection Model.** We projected the total execution time on a system with $P$ GPUs under an ideal model parallelism setup (modules are evenly distributed, communication overhead is neglected to isolate the update locking effect).

- The projected time for BP, dominated by its sequential critical path, is: $T_{BP}(P) = T_{BP\_fwd} + T_{BP\_bwd}$.

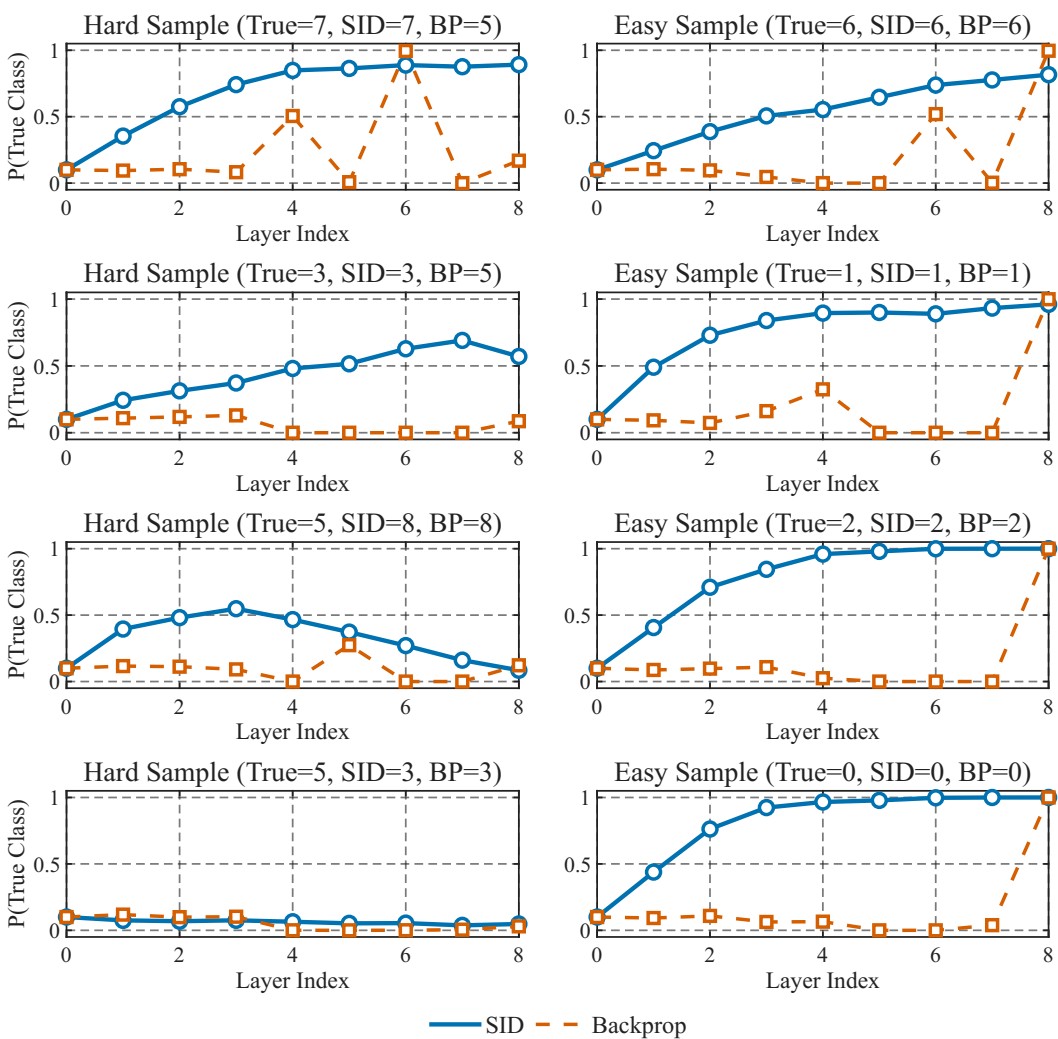

Figure 8: Full belief evolution trajectories for multiple hard (left columns) and easy (right columns) examples from the CIFAR-10 test set. SID's belief (blue) consistently shows a more stable monotonic progression compared to the potentially volatile trajectory of BP (orange).

- The projected time for SID is the sum of its sequential parts and the duration of its parallel phase: $T_{SID}(P) = T_{SID\_fwd\_teacher} + T_{SID\_fwd\_grad} + T_{parallel\_bwd}(P)$. The parallel backward time is determined by the busiest processor: $T_{parallel\_bwd}(P) = \max_{j=1...P}(\sum_{i \in \text{Modules on GPU}_j} T_{SID\_bwd\_i})$.

The speedup is then calculated as $Speedup(P) = T_{BP}(P)/T_{SID}(P)$. This model directly quantifies the architectural advantage of SID's parallelizable design.

## A  THEORETICAL ANALYSIS UNDER IMPERFECT OPTIMIZATION ($\epsilon$-ROBUSTNESS)

Proposition 2 in the main paper establishes a monotonic descent guarantee under the ideal condition that each module improves its local objective, i.e., $S_i(p_i) \leq S_i(p_{i-1})$. In practice, due to stochastic gradient descent and finite model capacity, modules may only find an approximate minimum. This section extends our analysis to this more realistic scenario.

We introduce the concept of an $\epsilon$-**improvement step**, where each module $f_i$ produces an output belief $p_i$ that satisfies:

$$S_i(p_i) \leq S_i(p_{i-1}) + \epsilon_i, \tag{7}$$

where $\epsilon_i \geq 0$ is the optimization error for module $i$. This error can arise from sources like gradient noise or terminating the optimization process early.

**Proposition 4** (Monotonic Descent with Bounded Error). *Suppose that each module's output $p_i$ satisfies the $\epsilon$-improvement condition $S_i(p_i) \leq S_i(p_{i-1}) + \epsilon_i$. Then, the KL divergence of the final belief $p_L$ to the target distribution $p_y$ is bounded as follows:*

$$D_{\mathrm{KL}}(p_L \| p_y) \leq D_{\mathrm{KL}}(p_0 \| p_y) - \frac{1-\alpha}{\alpha} \sum_{i=1}^{L} D_{\mathrm{KL}}(p_i \| p_{i-1}) + \frac{1}{\alpha} \sum_{i=1}^{L} \epsilon_i. \tag{8}$$

*This implies a simpler, more direct upper bound on the final performance degradation:*

$$D_{\mathrm{KL}}(p_L \| p_y) \leq D_{\mathrm{KL}}(p_0 \| p_y) + \frac{1}{\alpha} \sum_{i=1}^{L} \epsilon_i. \tag{9}$$

*Proof.* We start from the proof of Proposition 2 in Appendix A.2. The $\epsilon$-improvement condition is:

$$\alpha D_{\mathrm{KL}}(p_i \| p_y) + (1-\alpha) D_{\mathrm{KL}}(p_i \| p_{i-1}) \leq \alpha D_{\mathrm{KL}}(p_{i-1} \| p_y) + \epsilon_i.$$

Rearranging to isolate $D_{\mathrm{KL}}(p_i \| p_y)$:

$$\alpha D_{\mathrm{KL}}(p_i \| p_y) \leq \alpha D_{\mathrm{KL}}(p_{i-1} \| p_y) - (1-\alpha) D_{\mathrm{KL}}(p_i \| p_{i-1}) + \epsilon_i.$$

Dividing by $\alpha > 0$:

$$D_{\mathrm{KL}}(p_i \| p_y) \leq D_{\mathrm{KL}}(p_{i-1} \| p_y) - \frac{1-\alpha}{\alpha} D_{\mathrm{KL}}(p_i \| p_{i-1}) + \frac{\epsilon_i}{\alpha}.$$

Summing this inequality from $i = 1$ to $L$ yields a telescoping series on the left-hand side:

$$\sum_{i=1}^{L} (D_{\mathrm{KL}}(p_i \| p_y) - D_{\mathrm{KL}}(p_{i-1} \| p_y)) \leq \sum_{i=1}^{L} \left( -\frac{1-\alpha}{\alpha} D_{\mathrm{KL}}(p_i \| p_{i-1}) + \frac{\epsilon_i}{\alpha} \right).$$

$$D_{\mathrm{KL}}(p_L \| p_y) - D_{\mathrm{KL}}(p_0 \| p_y) \leq -\frac{1-\alpha}{\alpha} \sum_{i=1}^{L} D_{\mathrm{KL}}(p_i \| p_{i-1}) + \frac{1}{\alpha} \sum_{i=1}^{L} \epsilon_i.$$

Rearranging gives the first result. Since $D_{\mathrm{KL}}(p_i \| p_{i-1}) \geq 0$, we can drop the negative term to obtain the simpler upper bound:

$$D_{\mathrm{KL}}(p_L \| p_y) \leq D_{\mathrm{KL}}(p_0 \| p_y) + \frac{1}{\alpha} \sum_{i=1}^{L} \epsilon_i.$$

This completes the proof. $\square$

**Interpretation**: Proposition 4 provides a crucial robustness guarantee. It shows that the final prediction error is bounded by the initial error (from a uniform belief) plus the accumulated optimization errors from all modules, scaled by $1/\alpha$. As long as the local optimization is reasonably effective (i.e., $\epsilon_i$ are small), the overall learning process remains stable and does not diverge. This formalizes the intuition that SID is resilient to the imperfections inherent in practical training.

## B  ANALYSIS OF STALE TEACHER BELIEFS

The two-phase design of SID (Algorithm 1 and 2) involves generating teacher beliefs $p_{i-1}^{\mathrm{teacher}}$ using model parameters $\theta^{(t)}$ and then using these fixed teachers to compute gradients for updating to $\theta^{(t+1)}$. In asynchronous or parallel settings, these teachers might become "stale", meaning they were generated with slightly older parameters $\theta^{(t-k)}$ while the update happens on $\theta^{(t)}$. We analyze the impact of this staleness on the gradient computation.

Let $\theta$ denote the parameters of a module $f_i$, and let $p_{i-1} = f_{i-1}(\dots; \theta_{i-1}^{\text{old}})$ be the stale teacher belief generated with old parameters. The "fresh" teacher would be $p'_{i-1} = f_{i-1}(\dots; \theta_{i-1}^{\text{new}})$. The local loss for module $i$ is computed using the stale teacher:

$$\mathcal{L}_i(\theta_i) = \alpha D_{\text{KL}}(f_i(\dots)\|p_y) + (1-\alpha)D_{\text{KL}}(f_i(\dots)\|\text{sg}(p_{i-1})).$$

The true gradient should use $p'_{i-1}$. Let $g_i = \nabla_{\theta_i}\mathcal{L}_i$ be the computed gradient and $g'_i$ be the true gradient. The error is in the consistency term's gradient.

**Proposition 5** (Gradient Error Bound from Staleness). *Assume the module function $f_i$ is $L_f$-Lipschitz continuous with respect to its parameters, and the gradient of the KL divergence with respect to its second argument is $L_{\text{KL}}$-Lipschitz continuous in the relevant domain. The error in the computed gradient for module $i$ is bounded by:*

$$\|g_i - g'_i\| \le C \cdot \|\theta_{i-1}^{new} - \theta_{i-1}^{old}\|, \tag{10}$$

*for some constant $C$ that depends on $\alpha$, $L_f$, and $L_{\text{KL}}$.*

*Proof Sketch.* The difference between the gradients is:

$$\|g_i - g'_i\| = (1-\alpha)\|\nabla_{\theta_i}D_{\text{KL}}(f_i(\dots)\|p_{i-1}) - \nabla_{\theta_i}D_{\text{KL}}(f_i(\dots)\|p'_{i-1})\|.$$

Using the chain rule and Lipschitz continuity assumptions, this difference can be bounded by the difference in the teacher beliefs, $\|p_{i-1} - p'_{i-1}\|$. This belief difference is, in turn, bounded by the Lipschitz continuity of the predecessor module $f_{i-1}$:

$$\|p_{i-1} - p'_{i-1}\| = \|f_{i-1}(\dots; \theta_{i-1}^{\text{old}}) - f_{i-1}(\dots; \theta_{i-1}^{\text{new}})\| \le L_f \cdot \|\theta_{i-1}^{\text{old}} - \theta_{i-1}^{\text{new}}\|.$$

Combining these bounds yields the result. $\qquad\square$

**Interpretation**: The gradient error is directly proportional to the magnitude of parameter change between the time of teacher generation and its use. In standard training with small learning rates, $\|\theta^{\text{new}} - \theta^{\text{old}}\|$ is small, ensuring the gradient error is minimal. This analysis provides confidence that the two-phase approach is stable and the use of cached teachers is a valid and robust approximation.

## C  ADDITIONAL ABLATION STUDIES AND ROBUSTNESS ANALYSIS

To further assess the robustness of SID, we evaluate its performance under symmetric label noise, a challenging setting where a fraction of training labels are incorrect.

### C.1  ROBUSTNESS TO LABEL NOISE

**Experimental Setup**: We used the CIFAR-10 dataset and introduced symmetric label noise by randomly re-assigning the label of a certain percentage of training examples to one of the other 9 classes. We compared the final test accuracy of SID and standard BP after 100 epochs.

Table 5: Test accuracy (%) on CIFAR-10 with varying levels of symmetric label noise. SID demonstrates significantly higher robustness compared to standard backpropagation.

| Noise Level | 0% (Original) | 20% | 40% |
|---|---|---|---|
| Backpropagation (BP) | $84.32 \pm 0.45$ | $75.11 \pm 0.62$ | $58.24 \pm 0.88$ |
| SID (Ours) | $\mathbf{84.42 \pm 0.35}$ | $\mathbf{78.53 \pm 0.41}$ | $\mathbf{65.91 \pm 0.53}$ |
| Performance Gap ($\Delta$) | +0.10 | +3.42 | +7.67 |

**Analysis**: As shown in Table 5, while both methods perform similarly on the clean dataset, SID's performance advantage grows substantially as the level of label noise increases. We attribute this enhanced robustness to the local consistency term, $(1 - \alpha)D_{\text{KL}}(p_i\|\text{sg}(p_{i-1}))$. This term acts as a regularizer, forcing each module's update to be an incremental refinement of the previous module's belief. When the supervisory signal from the ground-truth label $p_y$ is noisy (incorrect), the consistency term provides a stable regularizing signal based on the consensus of earlier modules, preventing the model from aggressively overfitting to the incorrect label. BP, lacking such an explicit layer-wise regularizer, is more susceptible to fitting the noisy data.

