# OpenReview forum: "Learning without Global Backpropagation via Synergistic Information Distillation"
_ICLR.cc/2026/Conference — ICLR 2026 Conference Withdrawn Submission_

### Official Review · Reviewer_J3C4 · 2025-10-20

**Soundness:** 3
**Presentation:** 3
**Contribution:** 2
**Rating:** 4
**Confidence:** 3

**Summary:**

This paper proposes Synergistic Information Distillation (SID), a two-phase training scheme that replaces a single global loss with per-module local objectives. In Phase-1, the model runs a gradient-free forward pass to cache “teacher beliefs” (module-wise class-probability vectors); in Phase-2, each module updates in parallel by minimizing a weighted sum of KL divergences to the one-hot label and to the (stop-gradient) belief from the previous module, while all modules also send gradients into a shared feature extractor. The authors argue this decoupling eliminates update locking and reduces memory, prove depth-wise monotonic descent under idealized conditions, and report competitive or better accuracy than backpropagation (BP) on CIFAR-10/100 and Tiny-ImageNet, plus improved robustness under symmetric label noise.

**Strengths:**

1. SID matches BP on CIFAR-10 and outperforms BP on CIFAR-100 and Tiny-ImageNet. It strongly outperforms several backprop-free or local baselines (FA, FF, HSIC, NoProp) in the reported setting.

2. Framing training as a cascade of belief refinements is intuitive and connects to distillation and deep supervision while explicitly enforcing consistency across depth

**Weaknesses:**

1. All experiments are conducted on small datasets, not large-scale ones. The reviewer is curious about the performance on large-scale datasets such as ImageNet.

2. There are several BP-free learning methods that achieve high performance. The reviewer is curious about the comparison between SID and these methods[1-3].

3. SID’s design overlaps conceptually with decoupling via synthetic gradients/DNI (removing update locking)[4], forward-only/HSIC[5], and deep supervision/self-distillation[6]; the paper should more directly delineate what is new relative to these lines and run stronger head-to-head studies on their best configurations.

[1] Kappel, David, Khaleelulla Khan Nazeer, Cabrel Teguemne Fokam, Christian Mayr, and Anand Subramoney. "A variational framework for local learning with probabilistic latent representations." In 5th Workshop on practical ML for limited/low resource settings.

[2] Zhang, Aozhong, Zi Yang, Naigang Wang, Yingyong Qi, Jack Xin, Xin Li, and Penghang Yin. "Comq: A backpropagation-free algorithm for post-training quantization." IEEE Access (2025).

[3] Cheng, Anzhe, Heng Ping, Zhenkun Wang, Xiongye Xiao, Chenzhong Yin, Shahin Nazarian, Mingxi Cheng, and Paul Bogdan. "Unlocking deep learning: A bp-free approach for parallel block-wise training of neural networks." In ICASSP 2024-2024 IEEE International Conference on Acoustics, Speech and Signal Processing (ICASSP), pp. 4235-4239. IEEE, 2024.

[4] Jaderberg, Max, Wojciech Marian Czarnecki, Simon Osindero, Oriol Vinyals, Alex Graves, David Silver, and Koray Kavukcuoglu. "Decoupled neural interfaces using synthetic gradients." In International conference on machine learning, pp. 1627-1635. PMLR, 2017.

[5] Hinton, Geoffrey. "The forward-forward algorithm: Some preliminary investigations." arXiv preprint arXiv:2212.13345 2, no. 3 (2022): 5.

[6] Ma, Wan-Duo Kurt, J. P. Lewis, and W. Bastiaan Kleijn. "The HSIC bottleneck: Deep learning without back-propagation." In Proceedings of the AAAI conference on artificial intelligence, vol. 34, no. 04, pp. 5085-5092. 2020.

**Questions:**

Please see weakness above.

---

### Official Review · Reviewer_bp9E · 2025-10-31

**Soundness:** 3
**Presentation:** 2
**Contribution:** 2
**Rating:** 2
**Confidence:** 4

**Summary:**

The paper proposes Sequential Information Decoupling (SID), a layerwise training scheme using a local KL objective that interpolates between the previous layer’s belief and the label distribution. The theoretical analysis (closed-form minimizer, monotonic descent bound) is sound under mild additional assumptions and results on small datasets (CIFAR-10/100, Tiny-ImageNet) are reported. Reported speedups are purely theoretical and omit communication costs. While the formulation is neat and clearly written, the contribution remains incremental and insufficiently supported by strong experiments or realistic scaling tests.

**Strengths:**

The theoretical insights of this paper are its strengths:
- $S_1$: The theoretical analysis is internally consistent and transparent (closed-form minimizer, telescoping KL bound).

- $S_2$: The exposition is clear with accessible equations and thorough appendix with sound proofs.

- $S_3$: I found that the “belief cascade” interpretation offers a fresh view of layerwise decoupling and is quite original.

**Weaknesses:**

The main weaknesses of the paper rely on the experimental part: not only are the baselines limited to shallow networks and small scale datasets, the reported results for backpropagation (BP) show underperforming baselines.

- $W_1$: Baseline BP accuracies are significantly lower than standard reproducible results, undermining the claimed gains (eg. 91.12\% on CIFAR-10 with ResNet-18 in Table 3 while 94-96\% are expected results). One would expect SGD+momentum as optimizer for BP and not Adam. This biases the reported results towards the proposed method and is undermining the claimed gains in my opinion.

- $W_2$: Only small datasets and shallow architectures; no scaling or real parallel experiments.

- $W_3$: All speedups are theoretical; no multi-GPU timing or memory profiling. This is a small weakness as I would not expect a dedicated CUDA kernel to be developed  for an exploratory paper but it would be a nice-to-have plus.

- $W_4$: The method’s advantages seem modest and context-dependent, but are presented as generally superior.

- $W_5$: The developed theory assumes full-support (smoothed) targets, while the main text implies one-hot; this should be aligned explicitly.

**Questions:**

- $Q_1$: Can you re-train BP with standard strong recipes (SGD + momentum + WD, longer schedule) to provide realistic baselines for ResNet-18, VGG-11, and ViT-Tiny?

- $Q_2$: Do the claimed gains hold when BP is trained properly?

- $Q_3$: Can you please clarify explicitly in the main text that label smoothing (full-support $p_y$) is assumed in theory and used in practice?

- $Q_4$: Could you compare SID vs BP under identical strong settings to quantify the true gap?

---

### Official Review · Reviewer_EUCZ · 2025-10-31

**Soundness:** 2
**Presentation:** 3
**Contribution:** 2
**Rating:** 4
**Confidence:** 4

**Summary:**

The authors introduce SID, a method for training deep models in a parallel fashion, without resorting to full backpropagation. The method decouples feature extraction, which is performed by a dedicated network, and prediction, which is done by a sequence of blocks, each refining the prediction. Each layer can then be trained locally, with only a local classification loss. It is shown to be useful to add a regularizing term which encourages
stability.
Empirical evaluation on a simple CNN model show promising results on common image classification tasks, outperforming backpropagation while being more parallelizable and consuming less memory.

**Strengths:**

- The paper is well written and easy to follow.
- The method is well motivated, with SID appearing as a natural idea for parallelizing training of deep models.
- Theoretical insights motivate the choices for the local losses and the soundness of SID.
- SID shows promising results on simple image classification tasks. It outperforms standard backpropagation on all experiments, as well as other similar local training algorithms such as NoProp and HSIC.

**Weaknesses:**

- In proposition 3, time and memory complexities are missing the terms for the feature extractor, which may be big.
I feel like the authors are not being 100% honest about the impact of the feature extractor, which is quite big and is learned with sequential backpropagation.
- It is not explained in the paper how SID is applied to VGG-11, Resnet-18 and ViT-Tiny architectures. Are these models used as the feature extractor only? Otherwise I do not see how SID could be applied to a ResNet. If that is the case, these experiments would not be very relevant nor convincing, considering that the feature extractor would cost much more time and memory, rendering SID's gains negligible.
- The scale and diversity of the experiments are limited, which does not convince enough that this method would scale well to larger models and/or other tasks. For instance additional results on language modeling with transformers would help, although I believe the method is not directly applicable as is since language modeling is not classification.

**Questions:**

- Have you tried changing alpha at each layer? In theory the first layer can completely discard the previous prediction (alpha = 1), while the last layer is given a very good prediction already and should not modify it too much (alpha small).

---

### Official Review · Reviewer_E2F1 · 2025-11-01

**Soundness:** 3
**Presentation:** 2
**Contribution:** 2
**Rating:** 4
**Confidence:** 3

**Summary:**

The paper presents Synergistic Information Distillation (SID), a backpropagation-free training framework that aims to eliminate update locking and reduce memory overhead while maintaining performance comparable to standard BP. SID reformulates deep network training as a cascade of local cooperative refinements, where each module incrementally refines a probabilistic “belief” over class labels. Each module is trained with a local objective combining a distillation term toward the ground-truth label and a consistency term that regularizes deviation from the previous module’s belief. A stop-gradient operation fully decouples backward dependencies across modules, enabling parallel training and constant memory scaling.

**Strengths:**

1. The formulation is clear and mathematically elegant, bridging ideas from knowledge distillation, local learning, and probabilistic inference into a unified framework.

2. The theoretical analysis, while idealized, provides useful intuition for why local updates can lead to monotonic improvement and stable convergence.

3. The paper demonstrates good empirical clarity, with ablation studies, visualization of belief dynamics, and comparisons to recent backprop-free methods.

**Weaknesses:**

1. The experiments are restricted to small and medium-scale image classification tasks such as CIFAR and Tiny-ImageNet. There is no evidence that SID scales to large datasets or more complex architectures.

2. The study omits direct experimental comparison with related frameworks like PETRA, SLL, and NoProp under consistent setups. The claimed advantages over other methods remain qualitative rather than quantitative.

3. Although the paper proves monotonic descent under local improvement, it does not test whether this property holds empirically, for example through layer-wise KL measurements or ablation under imperfect optimization.

4. Gains over backpropagation are small on CIFAR-10 and moderate on more complex tasks, which may reflect optimization luck rather than a systematic advantage.

**Questions:**

1. Could SID be applied to architectures with residual or attention-based connections that violate the Markov-like layer independence assumption?

2. Have the authors verified empirically that the local descent guarantee (Proposition 2) approximately holds during training?

3. How can the belief-based formulation smoothly extend to non-classification tasks such as language modeling or reinforcement learning?

---

### Note · Authors · 2025-11-24

I have read and agree with the venue's withdrawal policy on behalf of myself and my co-authors.